# Overcoming Redundant Context in Auto-Regressive LLMs with Dynamic Draft Refinement

## Abstract

Large language models (LLMs) have demonstrated strong reasoning abilities, driven in part by reinforcement learning–based optimization methods such as Reinforcement Learning with Verifiable Rewards (RLVR). These methods encourage a slow thinking paradigm, where models produce detailed intermediate steps between designated "thinking tokens." Models employing such strategies are commonly referred to as Large Reasoning Models (LRMs). Despite notable progress, LRMs lack selective retention—the ability to discard redundant reasoning while preserving only the intermediate results that are useful for subsequent steps. In contrast, humans dynamically maintain a structured "working state", continuously filtering out unproductive thoughts and retaining essential ones. To examine this limitation, we introduce a lightweight arithmetic benchmark designed to isolate reasoning behaviors. Through systematic evaluation, we show that redundant intermediate traces—specifically those that are low-quality or irrelevant—significantly degrade performance.. Analyses of task accuracy, behavioral patterns, and attention allocation confirm that LRMs often rely on the entire reasoning context, including outdated or unhelpful information. To address this issue, we propose Dynamic Context Refinement (D-Refine), an inference-time mechanism that selectively organizes and condenses reasoning steps as they are generated. Experiments on diverse benchmarks demonstrate consistent performance gains, highlighting the importance of maintaining a well-structured working state for accurate and efficient reasoning. This work establishes selective retention as a key principle for improving LRM reasoning.

## 1 Introduction

Large language models (LLMs) have become a transformative force in natural language processing, achieving strong performance on tasks such as code generation, multi-turn dialogue, text summarization, and mathematical problem solving (Minaee et al., 2024; Zhao et al., 2023). A central factor in this success is improved reasoning ability; recent advances targeting reasoning have substantially raised performance on diverse benchmarks. Reinforcement learning (RL)-based optimization has been a major contributor. The exceptional reasoning demonstrated by ChatGPT (hereafter O1) highlighted RL's potential for enhancing model reasoning (Jaech et al., 2024). Building on this, the Reinforcement Learning with Verifiable Rewards (RLVR) framework was introduced, and one prominent open-source implementation, R1, achieves reasoning performance comparable to O1 (Shao et al., 2024; Guo et al., 2025). RLVR facilitates a "slow thinking" paradigm in which models emit detailed intermediate outputs between designated "thinking tokens (<think>)." Models that adopt these reasoning-enhancement strategies are commonly referred to as Large Reasoning Models (LRMs), and this paradigm has led to marked improvements on standard reasoning benchmarks.

A key feature of the "slow thinking" process in LRMs is the generation of textual outputs between designated "thinking tokens," which capture diverse reasoning behaviors such as problem decomposition, hypothesis formation, backtracking, directional shifts, and the summarization of intermediate results(Yao et al., 2023; Bilal et al., 2025). In many open-source reinforcement learning frameworks, these behaviors occur within explicit reasoning segments marked by special tokens that signal the beginning and end of a reasoning phase (Guo et al., 2025). After each phase, the model consolidates

its internal trace into a concise summary that informs the final answer. This mechanism parallels human problem solving: individuals jot down exploratory thoughts, discard invalid hypotheses, and progressively synthesize structured responses. Similarly, students solving exam problems often rely on draft notes before presenting a polished solution. We refer to this process as the *draft-summarize paradigm*.

Motivated by this analogy, recent studies have sought to enhance reasoning efficiency and accuracy through techniques such as prompt-based reasoning instructions, inference-time probability adjustments (Wang et al., 2025b; Muennighoff et al., 2025), improved memory utilization across subtasks, and refined reward functions via reward shaping (Xiang et al., 2025a; Wang et al., 2025a; Liu et al., 2025). These methods have achieved notable improvements on standard benchmarks.

To investigate whether LRMs should rely on all previously generated intermediate reasoning within the draft-summarize paradigm, we constructed scenarios where contexts contained redundant traces. This allowed us to examine their impact on subsequent reasoning steps. To ensure a focused evaluation, we developed a lightweight arithmetic benchmark with two-decimal addition and subtraction problems, minimizing memorization effects and emphasizing reasoning behavior. Our experiments analyzed the influence of intermediate reasoning traces on final performance from multiple perspectives, including behavioral patterns, attention allocation, and end-task accuracy.

However, despite these advancements, current methods still lack a critical aspect of human reasoning: selective retention. When solving complex problems, humans rarely rely on their entire thought history. Instead, they maintain a dynamic "working state," discarding unproductive reasoning while preserving results essential for subsequent steps. For example, in graph coloring, once a strategy fails, humans simply mark it invalid and move on, rather than revisiting its details. In contrast, autoregressive models attend indiscriminately to the entire reasoning context (Vaswani et al., 2017), including outdated or irrelevant traces, which can introduce noise and hinder accurate predictions(Ghosal et al., 2025; Kuo et al., 2025). This raises a fundamental question: should LRMs reason over their full history of thoughts, or selectively retain only what is useful?

These analyses reveal a key limitation of current LRMs: the lack of selective retention—the ability to discard redundant traces while preserving useful conclusions—which can reduce accuracy by up to 20%. To address this bottleneck, we propose **Dynamic Context Refinement (D-Refine)**, an inference-time framework that dynamically organizes and condenses reasoning steps as they are generated. Empirical evaluations on diverse benchmarks show that D-Refine consistently improves performance, demonstrating that maintaining an organized intermediate "reasoning state" is critical for efficient and accurate multi-step reasoning. By both revealing this limitation and providing an effective solution, our work establishes selective retention as a central component for advancing the reasoning capabilities of LRMs.

## 2 RELATED WORKS

### 2.1 MEMORY-AUGMENTED METHODS FOR ENHANCED REASONING

Organizing and leveraging prior reasoning experiences derived from similar task inputs has become a common approach to enhance the problem-solving abilities of LLMs. Reflexion (Shinn et al., 2023) introduced explicit reflection on failed trajectories, treating unsuccessful attempts as explicit memory for guiding future attempts. EXPEL (Zhao et al., 2024) builds on this idea by collecting cross-task experiences into a structured pool for reuse. MSI-Agent (Fu et al., 2024) extends this by transforming trajectory-level experiences into hierarchical long-term memory. SelfGoal (Yang et al., 2025b) investigates selective retrieval of intermediate reasoning traces, while Agent Workflow Memory (Wang et al.) focuses on structuring task-specific execution states, though mainly in procedural domains.

These methods highlight the benefits of external memory but largely concentrate on reusing experiences across tasks or episodes. In contrast, we study an underexplored direction: how draft–summary style reasoning within a single reasoning episode shapes downstream inference. We further propose a context-refinement mechanism that selectively organizes intermediate traces, thereby improving draft–summary style reasoning.

## 2.2 DRAFT–SUMMARY REASONING UNDER RL TRAINING

RL has been central to improving the reasoning abilities of LLMs. Early methods such as RLHF (Ouyang et al., 2022; Peng et al., 2023) aligned models with human preferences, and later studies showed that RL-based fine-tuning can also enhance logical reasoning (Wang et al., 2024). Alternatives like DPO (Rafailov et al., 2023; Zhang et al., 2024) further demonstrated that preference-guided supervision with curated trajectories benefits reasoning.

A major shift came with the *slow thinking* paradigm in O1 (Jaech et al., 2024), where models generated intermediate drafts before synthesizing final answers. DeepSeek-R1 (Shao et al., 2024; Guo et al., 2025) advanced this idea through RLVR, explicitly guiding models to explore reasoning paths and consolidate them into concise outputs. This style of problem solving—drafting exploratory traces followed by summarization—is what we refer to as the *draft–summary paradigm*. Subsequent models, including Qwen3 (Yang et al., 2025a) and Magistral (Rastogi et al., 2025), have adopted similar strategies, establishing RL as a key driver of structured reasoning.

## 2.3 IMPROVING THE EFFICIENCY OF LONG-FORM REASONING

Building on draft–summary LRMs, recent work has shifted toward improving the efficiency and reliability of long-form reasoning. Analyses such as Underthinking (Wang et al., 2025b) and Overthinking (Chen et al., 2024) attribute reasoning failures to premature validation or unnecessary elaboration, and propose probability-level interventions to better regulate generation. S1 (Muennighoff et al., 2025) introduces explicit budget control by embedding token-level constraints, while meta-reasoning frameworks (Xiang et al., 2025b; Wan et al., 2025; Bilal et al., 2025) restructure reasoning into multi-agent or stepwise processes for greater interpretability. Complementary approaches such as procedural reward shaping (Qu et al., 2025) further guide models toward more consistent reasoning paths. Despite these advances, one critical gap remains: current methods pay little attention to how intermediate drafts themselves shape subsequent reasoning. Our work addresses this gap by showing that unfiltered or poorly structured drafts can mislead inference, and by introducing a context-refinement mechanism that selectively organizes intermediate reasoning traces. This design enables LRMs to maintain a well-structured intermediate reasoning state, ultimately leading to more efficient and accurate reasoning.

# 3 PRELIMINARY ANALYSIS: THE ROLE OF CONTEXT IN STEP-BY-STEP REASONING

In autoregressive LRMs, each step of *slow thinking* is conditioned on previously generated intermediate drafts. However, Unlike humans, who update their working memory dynamically and selectively retain only task-relevant conclusions (Oberauer, 2002), current LLMs indiscriminately attend to the entire draft history. Our central hypothesis is that irrelevant or low-quality drafting context can mislead subsequent reasoning, ultimately degrading overall performance. To investigate this issue, we conduct a preliminary analysis from three perspectives: (i) behavioral patterns of reasoning, (ii) end-task accuracy, and (iii) attention allocation in the final layer.

## 3.1 TASK FORMULATION

To ground our analysis, we first recall the standard autoregressive formulation of language modeling. Given a sequence $X = (x_1, \ldots, x_n)$, the probability of generation is factorized as:

$$P(X; \theta) = \prod_{i=1}^{n} P(x_i \mid x_{<i}; \theta), \quad (1)$$

where $x_{<i}$ denotes the preceding tokens.

To capture step-by-step reasoning, we adopt a **draft–summary paradigm**. Given a question $Q$, the model first produces a draft $S$ and then generates a final answer $A$, yielding:

$$P(S, A \mid Q) = \left( \prod_{j=1}^{m} P(s_j \mid Q, s_{<j}) \right) \cdot P(A \mid Q, S), \quad (2)$$

where $S = (s_1, \ldots, s_m)$ is an ordered sequence of draft segments. This formulation explicitly separates the drafting and summarization phases, aligning with the "slow thinking" process in Section 1.

## 3.2 EXPERIMENTAL SETTING

We evaluate a representative open-source LRM, QwQ-32B (Team, 2025b), under conditions where in-context drafts are deliberately irrelevant or of low quality. If models are still disrupted by such context, despite its lack of task relevance, this reveals an inherent weakness: current LRMs struggle to selectively attend to useful information during multi-step reasoning. In more realistic scenarios, where draft utility is not explicitly annotated, this limitation suggests that misleading context may continue to degrade downstream reasoning quality.

To minimize potential data leakage from existing reasoning benchmarks during pretraining (Wu et al., 2025), we construct a new arithmetic (**Arith**) benchmark based on two-decimal addition and subtraction. The benchmark controls complexity by varying the depth of nested parentheses and the number of operations per level. Although these problems are straightforward for humans, they are designed to evaluate reasoning robustness rather than absolute capability. Two factors support the validity of this benchmark: (i) the problems are novel and unlikely to overlap with pretraining corpora, thereby emphasizing reasoning over memorization; and (ii) performance degrades substantially as complexity increases (e.g., QwQ-32B drops from nearly 100% to around 50%), providing a clear signal for distinguishing models' robustness. The complexity of the arithmetic benchmark and QwQ-32B's performance are shown in Figure 1 (a). An example problem from the benchmark is provided in 1 (b). Details of the Arith benchmark construction are provided in Appendix A.1.

Irrelevant or low-quality drafts are introduced by using a two-question input format. The second question is always drawn from our arithmetic benchmark, while the first is selected from two types of questions identified by QwQ-32B's performance:

- **Solvable questions (SQ)**, which QwQ-32B answers correctly, producing coherent drafts;
- **Unsolvable questions (UQ)**, which QwQ-32B fails to solve but still generates a full reasoning trace, yielding logically flawed drafts.

These settings simulate contexts of different quality. To control for length effects, we use a single proof-style problem from graph theory to populate the first question slot in both cases. Further details are provided in Appendix A.2.2.

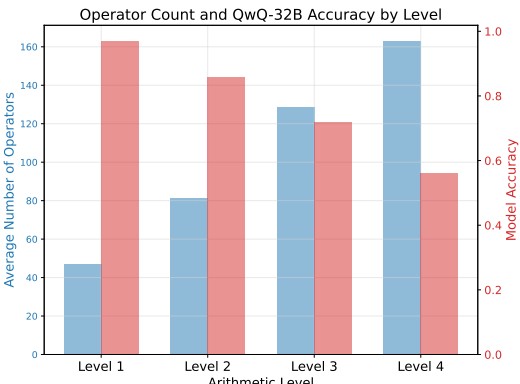

(a) Operator Count and QwQ-32B Accuracy by Level  (b) Case from arithmetic benchmark (level 1)

Figure 1: (a) shows the average number of operators per problem (blue bars) across the four levels of the benchmark, with QwQ-32B's accuracy on the same problems shown in red. (b) presents an example problem from the arithmetic benchmark.

## 3.3 OVERALL PERFORMANCE

When LRMs are provided with two questions as a combined input prompt, we refer to them as *Problem 1* and *Problem 2*. LRMs typically follow a sequential strategy: they first attempt *Problem 1*

and then shift—often producing drafting phrases such as "Now turn to Problem 2"—to address the second problem. Our evaluation focuses exclusively on accuracy for the arithmetic benchmark (*Problem 2*). We do not assess accuracy on *Problem 1*, as it primarily serves to generate fluent but semantically diverse drafting contexts of varying difficulty levels, which are not directly related to the subsequent reasoning task. As shown in Table 1, QwQ-32B exhibits a accuracy degradation under the combined input setting. When *Problem 1* is unsolvable, accuracy decreases by roughly 15%; even when *Problem 1* is solvable, accuracy still declines by approximately 4%. These results indicate that prior reasoning traces, even if logically coherent, can interfere with subsequent problem solving, highlighting the nontrivial effects of context interference.

To disentangle the effects of context length and semantic quality, we conducted a controlled ablation study. Fixed-length contexts were created by truncating or padding the reasoning trace from *Problem 1* (details in Appendix A.2). Across all lengths, arithmetic accuracy following unsolvable drafts remained up to 10% lower than that following solvable drafts. This demonstrates that the semantic quality of intermediate reasoning—not just its length—plays a critical role in shaping reasoning robustness.

### 3.4 BEHAVIORAL PERFORMANCE

To further investigate the source of the observed accuracy gap, we analyze the reasoning behavior of QwQ-32B. Following prior work on reasoning efficiency (Wang et al., 2025b), we track behavioral signals such as the occurrence of reflective or corrective phrases (in our study, we focus on "alternatively") to quantify how often the model shifts strategies or backtracks when solving the second problem. While not exhaustive, this signal is sufficient to capture the degree of deliberation in the model's reasoning. We evaluate the model under the input conditions described in Table 1, measuring both the **rate** (proportion of samples containing "alternatively") and the **freq**uency (average number of occurrences per sample) of this token during *Problem 2* reasoning. As shown in Table 2, a clear pattern emerges: irrelevant context reduces reflective behavior. Both the probability and frequency of reflective tokens decline when *Problem 1* is unrelated to the arithmetic benchmark. This effect is particularly pronounced when *Problem 1* is unsolvable, compared to when it is solvable, despite both being unrelated to the target task. These results indicate that lower-quality drafting context increases the likelihood of underthinking, thereby diminishing the robustness of the model's reasoning process.

### 3.5 ATTENTION LAYER OBSERVATION

The preceding analyses indicate that irrelevant or low-quality drafting context degrades downstream reasoning. To understand why, we examine the model's attention during *Problem 2*. Humans solving sequential tasks typically suppress earlier redundant context and focus on the current subtask, revisiting prior information only when necessary. In contrast, our results show that LRMs continue to allocate substantial attention to *Problem 1* even while addressing the arithmetic task.

Specifically, once *Problem 2* reasoning begins, we measure aggregated attention in the final self-attention layer toward three segments: the problem statement, *Problem 1*'s drafting trace, and *Problem 2*'s drafting trace (Figure 2). Using 15 sampled responses from both SQ+Arith and UQ+Arith across four difficulty levels, we compute attention distributions over the first 3k tokens of *Problem 2* drafting (considering only cases where *Problem 2* exceeds 3k tokens). We find that, although attention to *Problem 1* gradually decreases, it remains as high as 0.2 at the 3k-th token without external intervention (truncation or padding), indicating that the model fails to adequately suppress redundant prior context, thereby undermining robustness in multi-step reasoning.

## 4 METHODOLOGY

As demonstrated in Section 3 , a central limitation of current LRMs is their inability to filter out redundant information from accumulated drafts. Under controlled settings, we showed that redundant or low-quality context degrades downstream reasoning, as evidenced by reduced accuracy and persistent attention to unhelpful content. This weakness aligns with prior observations that vanilla LRMs often exhibit redundancy in their reasoning processes (Yue et al., 2025), where longer contexts do not necessarily improve performance and instead increase backtracking.

Table 1: Arithmetic performance of QwQ-32B under the combined question input. **Acc** denotes accuracy, and **CL** (tokens) indicates the context length (number of tokens) present when starting to reason on Problem 2.

| Input (QwQ -32B) | Level 1 | | Level 2 | | Level 3 | | Level 4 | |
|---|---|---|---|---|---|---|---|---|
| | Acc | CL | Acc | CL | Acc | CL | Acc | CL |
| Arith | 0.97 | 226 | 0.86 | 373 | 0.72 | 583 | 0.56 | 734 |
| SQ + Arith | 0.95 | 7186 | 0.85 | 6980 | 0.71 | 7062 | 0.43 | 7595 |
| UQ + Arith | 0.90 | 15414 | 0.64 | 15458 | 0.49 | 15046 | 0.36 | 14637 |
| Ablation on context length | | | | | | | | |
| SQ + Arith w Pad | 0.95 | 15515 | 0.88 | 15522 | 0.68 | 15112 | 0.41 | 14758 |
| UQ + Arith w Truncate | 0.90 | 6928 | 0.71 | 6643 | 0.62 | 6732 | 0.36 | 7306 |

Table 2: Behavior-level performance of QwQ-32B. We report the **rate** and **freq**uency of the reflective token "alternatively" during *Problem 2* reasoning under different input settings.

| Input (QwQ -32B) | Level 1 | | Level 2 | | Level 3 | | Level 4 | |
|---|---|---|---|---|---|---|---|---|
| | Rate | Freq | Rate | Freq | Rate | Freq | Rate | Freq |
| Arith | 0.99 | 5.73 | 0.98 | 6.46 | 0.94 | 7.79 | 0.95 | 8.24 |
| SQ + Arith | 0.86 | 3.17 | 0.84 | 3.34 | 0.86 | 3.39 | 0.93 | 5.57 |
| UQ + Arith | 0.26 | 0.72 | 0.46 | 1.21 | 0.53 | 2.36 | 0.65 | 8.8 |
| Ablation on context length | | | | | | | | |
| SQ + Arith w Pad | 0.69 | 2.09 | 0.66 | 2.24 | 0.81 | 4.34 | 0.86 | 5.35 |
| UQ + Arith w Truncate | 0.56 | 2.55 | 0.61 | 6.55 | 0.52 | 2.05 | 0.62 | 4.17 |

Existing approaches attempt to address this issue by constraining generation—via prompt engineering, model-level control, or hidden-state interventions—to favor useful reasoning steps. However, the usefulness of a step is often revealed only *after* it is generated, and effective reasoning typically requires exploring suboptimal paths and backtracking.

Motivated by this limitation, we introduce **D**ynamic Context **Refine**ment. Rather than constraining draft generation in advance, our method dynamically refines the accumulated context during reasoning. By selectively pruning or condensing intermediate drafts, it ensures that each subsequent step is conditioned only on the most relevant and high-quality information. This design mirrors human cognition, where refinement of prior thoughts are essential for clarity and progress. As a result, our approach prevents performance degradation from redundant context and enables more robust and efficient multi-step reasoning.

## 4.1 THE D-REFINE FRAMEWORK

We introduce **D-Refine**, a dynamic reasoning framework designed to enhance multi-step drafting by continuously refining the intermediate reasoning state. As illustrated in Figure 3, D-Refine ensures that each reasoning step is conditioned on a concise, high-quality context, mitigating the negative impact of redundant information. The details of D-Refine are presented in Algorithm 2.

**1. Triggering the Refinement Phase**    The refinement phase is initiated based on a length-based criterion. Let $s_{\text{len}}$ denote a predefined segment length. Once the number of newly generated tokens since the last refinement (or since the start of reasoning) reaches $s_{\text{len}}$, and the LRM has not yet terminated the draft (e.g., by producing a final answer token '</think>'), generation is temporarily paused. The newly generated segment $\Delta S = (s_{t_r+1}, \ldots, s_t)$ is then extracted for refinement.

**2. The Refinement Mechanism**    Refinement operates on the combination of previously refined context $\hat{S}_r$ and the new segment $\Delta S$, ensuring that historical reasoning is continuously integrated:

$$\hat{S}_{r+1} = f_{\text{refine}}(\hat{S}_r \cup \Delta S), \tag{3}$$

where $f_{\text{refine}}(\cdot)$ is implemented via prompting-based summarization and pruning:

- **Summarization:** Condenses the combined reasoning steps into a concise, structured note that captures essential deductions.

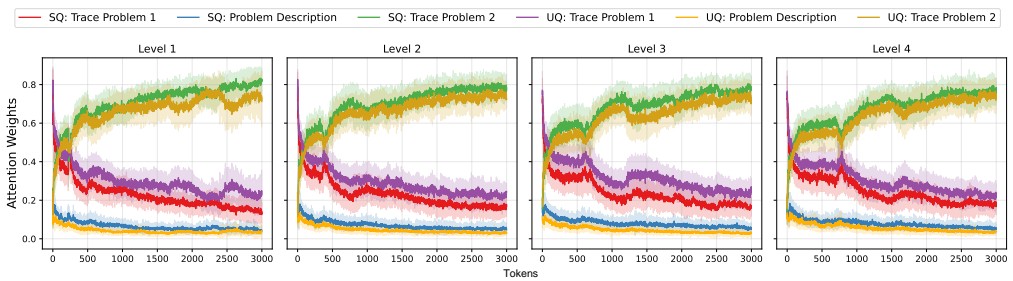

Figure 2: Cumulative attention weights of the LLM during reasoning, computed separately for UQ and SQ inputs, over three context segments: the problem statement, Problem 1 drafting trace, and Problem 2 drafting trace.

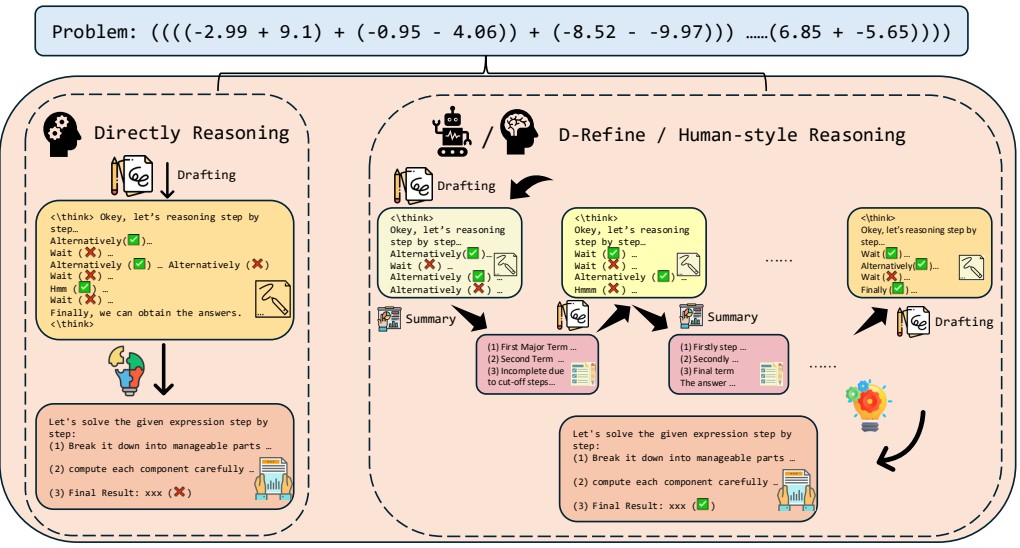

Figure 3: Framework of D-Refine. Unlike Direct Reasoning, D-Refine incorporates refinement of the existing context (yellow blocks) within each reasoning step, yielding a refined, high-quality context (pink blocks) that enables more robust reasoning.

- **Pruning:** Removes irrelevant, low- quality, or counterproductive content that could degrade subsequent reasoning.

The output $\hat{S}_{r+1}$ forms a refined context that integrates all previous reasoning while maintaining focus and clarity.

**3. Resuming the Drafting Process** Following refinement, the LRM continues generating the next segment conditioned on both the original problem $Q$ and the updated context $\hat{S}_{r+1}$:

$$s_{t+1:t+s_{\text{len}}} \sim P(s \mid Q, \hat{S}_{r+1}). \tag{4}$$

This process repeats iteratively: after generating a new segment, it is merged with the existing refined context and passed through $f_{\text{refine}}$ before generating the next tokens. Formally, for the $(r+1)$-th refinement cycle:

$$\hat{S}_{r+2} = f_{\text{refine}}(\hat{S}_{r+1} \cup s_{t+1:t+s_{\text{len}}}), \tag{5}$$

guaranteeing that the model maintains a coherent, relevant working state throughout multi-step reasoning.

By continuously consolidating the reasoning trace in this manner, D-Refine emulates human cognitive strategies, where intermediate thoughts are periodically summarized and organized to guide effective

next steps. This dynamic integration of past and newly generated reasoning ensures robustness and efficiency, allowing the LRM to focus on high-utility steps rather than being distracted by verbose or low-quality drafts.

## 5 EXPERIMENTS

### 5.1 SETUP

We conduct extensive experiments on both synthetic and standard reasoning benchmarks to evaluate D-Refine. Our study aims to address three key research questions: (1) How well does D-Refine mitigate the negative impact of low-quality or irrelevant drafting context, as motivated in Section 3.2? (2) How well does D-Refine generalize across different models and benchmarks? (3) How sensitive is D-Refine to hyperparameter choices?

**Base models.** We select three LRMs that already exhibit slow-thinking capabilities as our base models. To ensure diversity in both source and model scale, we consider several representative open-source reasoning models: `DeepSeek-Distill-Llama-8B`(DeepSeek-AI, 2025), `Qwen3-4B-Thinking` (Team, 2025a), and `QwQ-32B` (Team, 2025b).

**Baseline & Benchmarks** We compare D-Refine against *direct reasoning*. Unlike prior work, D-Refine takes a novel yet simple perspective: improving LRM reasoning by suppressing the negative influence of redundant context. Our goal is not to propose a standalone reasoning algorithm but rather to demonstrate the feasibility of this direction. Importantly, D-Refine can be integrated into any draft-based reasoning approach. Beyond our constructed two-decimal arithmetic benchmark, we evaluate D-Refine in more general settings, including single-question inputs, we adopt widely used benchmarks such as `MATH500`(Lightman et al., 2023), `AIME'24`, and `AIME'25`.

**Experimental details.** For reproducibility, we fix decoding hyperparameters to `temperature=0`. Although this may not yield optimal performance for every model, using a unified configuration enables fair comparison. Direct reasoning is allowed up to 64k tokens to prevent premature termination from insufficient context. For D-Refine, each refinement phase processes up to 32k tokens. After the maximum number of refinement (3) is reached, the model can generate up to 32k tokens to complete reasoning. The refinement is triggered every 9k tokens of newly generated content.

### 5.2 RQ1: EFFECTIVENESS UNDER REDUNDANT CONTEXT

Table 3 reports results in the controlled setting described in Section 3.2, where the drafting trace of *Problem 1* is irrelevant to *Problem 2*. In this setting, we apply a single refinement pass over *Problem 1*'s trace before continuing reasoning for *Problem 2*. Importantly, the correctness of *Problem 1* remains unaffected—the gains arise solely from restructuring noisy drafts into a more organized context. Compared to direct reasoning, D-Refine achieves a substantial improvement of 6–25% in arithmetic accuracy under the *UQ + Arith* setting, where *Problem 1* produces logically flawed drafts.

Table 3: Accuracy of Direct Reasoning (DR) and D-Refine under the SQ (UQ) + Arith setting, where redundant context is explicitly identifiable.

| Level | SQ + Arith | | UQ + Arith | | Arith |
|---|---|---|---|---|---|
| | DR | D-Refine | DR | D-Refine | D-Refine |
| 1 | 0.95 | 0.94 | 0.90 | 0.96 | **0.98** |
| 2 | 0.85 | 0.87 | 0.64 | 0.85 | **0.91** |
| 3 | 0.71 | 0.69 | 0.49 | 0.74 | **0.76** |
| 4 | 0.43 | 0.48 | 0.36 | 0.57 | **0.60** |

flawed drafts. By contrast, under the *SQ + Arith* setting, where drafts are already coherent, the average improvement is only about 1%. This asymmetry highlights that refinement brings the greatest benefit when drafts are of low quality, further underscoring context quality as a key factor influencing downstream reasoning performance. Due to space constraints, we present the output of refinement in Appendix A.4, while the full examples (mostly exceeding 10k tokens) are provided in the supplementary materials.

Table 4: Arithmetic performance of QwQ-32B under the combined question input. **Acc** denotes accuracy, and **CL** (tokens) indicates the context length (number of tokens) present when starting to reason on Problem 2.

| Model | AIME'24 | | AIME'25 | | MATH500 | | Avg Acc | |
|---|---|---|---|---|---|---|---|---|
| | DR | D-Refine | DR | D-Refine | DR | D-Refine | DR | D-Refine |
| QwQ-32B | 0.733 | 0.733 | **0.700** | 0.667 | 0.974 | **0.982** | **0.802** | 0.794 |
| Qwen3-4B | **0.800** | 0.767 | 0.767 | **0.800** | 0.984 | **0.988** | 0.850 | **0.852** |
| Llama-8B | 0.233 | **0.400** | 0.167 | **0.267** | 0.792 | **0.852** | 0.397 | **0.506** |

## 5.3 RQ2: GENERALIZATION ACROSS BENCHMARKS

In the controlled setting of Section 3.2, irrelevant context can be explicitly separated from useful reasoning traces. In more general scenarios, however, such distinctions are less clear. To evaluate the broader effectiveness of D-Refine, we trigger refinement based on token length, selecting the preceding semantically complete word (e.g., "alternative", "wait") as the boundary. We first evaluate on the Arith benchmark, where D-Refine achieves an average improvement of 3.8% over direct reasoning (Table 3, "Arith" column), indicating that the method can enhance stability in multi-step arithmetic reasoning.

Beyond this constructed setting, we further test on widely used mathematical reasoning benchmarks. As shown in Table 4, D-Refine yields consistent gains across models and datasets, with nearly 10% improvement on AIME for DeepSeek-Distill-Llama, and 2–3% improvements in most other settings. Notably, on the simpler MATH500 benchmark, while the absolute accuracy improvement is smaller, the relative reduction in error rate exceeds 25% across all three models. This demonstrates that D-Refine not only improves accuracy but also significantly reduces the likelihood of errors, highlighting its potential to enhance robustness in general reasoning tasks.

## 5.4 RQ3: SENSITIVITY TO HYPERPARAMETERS

We ablate two key hyperparameters: the number of refinement steps {2,3} and the refinement trigger length {3000, 6000, 9000}. As shown in Figure 4, D-Refine achieves the most stable performance under the (3, 9000) setting. While the effect of trigger length is less consistent, larger values generally help—supporting the intuition that although D-Refine may disrupt fluency, pruning redundant history compensates for this disruption. This indicates that managing larger context can stabilize reasoning. Furthermore, when each refinement brings positive utility, increasing the number of refinement tends to yield additional gains. Still, how to identify the optimal hyperparameters for each model remains an open and worthwhile direction.

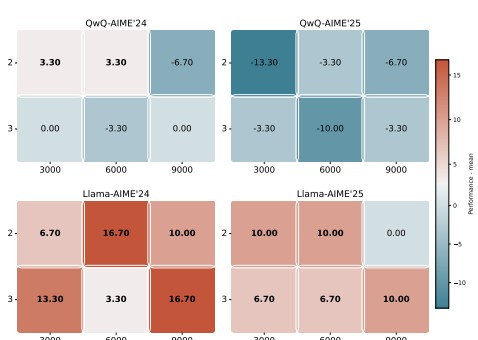

Figure 4: Performance differences between D-Refine and Direct Reasoning across hyperparameters. Red denotes gains, blue denotes drops, with color intensity indicating magnitude.

## 6 CONCLUSION

This work focuses on the negative impact of irrelevant or low-quality context on the robustness of multi-step reasoning, and demonstrates through experiments that autoregressive LRMs have limitations in mitigating such interference. To address this, we propose a lightweight method, D-Refine, which alleviates distractions by structurally organizing intermediate reasoning drafts. Results across multiple benchmarks show that D-Refine effectively enhances reasoning performance. Our findings highlight the importance of managing generated context during reasoning and open new directions for exploring adaptive reasoning optimization methods.

## 7 ETHICS STATEMENT

Our work relies primarily on open-source or synthetically generated datasets in the mathematical domain, and all employed large language models are also open-source. As the study is conducted purely for research purposes, we do not anticipate ethical concerns.

## 8 REPRODUCIBILITY STATEMENT

To ensure the reproducibility of our experimental results, we set the decoding temperature to 0 and report all relevant inference hyperparameters in the main text. Additionally, the full specifications of system and user prompts used for LLM reasoning are provided in the appendix. As our work focuses on analyzing reasoning behaviors and introducing a method for the inference stage, all experiments can be fully reproduced given sufficient computational resources.

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

# A APPENDIX

## A.1 ARITHMETIC BENCHMARK

### A.1.1 PROBLEM GENERATION

We implemented a function for generating arithmetic problems, primarily using recursion and random value generation. The general procedure for constructing a problem is outlined in Algorithm 1. The generator recursively constructs arithmetic expressions by randomly selecting two-decimal numbers or binary operators (+, -). At each recursive step, the algorithm decides whether to terminate and return a number (based on current depth and a probabilistic criterion, Line 1–3) or continue building a nested expression by generating left and right sub-expressions (Line 4–6). The complexity of generated problems can be controlled by the min_depth and max_depth, which set the minimum and maximum recursion depths, respectively.

---

**Algorithm 1** Recursive Problem Generator

---

**Input:** Current depth `depth`, minimum depth `min_depth`, maximum depth `max_depth`
**Output:** Generated problem `prob`

1 **if** `depth ≥ max_depth OR (depth ≥ min_depth AND random() < 0.3)` **then**
2    | num ← random two-decimal number;
3    | **return** num

4 operator ← random choice from $\{+, -\}$
5 left_expr ← Recursive_Problem_Generator(depth+1, min_depth, max_depth)
6 right_expr ← Recursive_Problem_Generator(depth+1, min_depth, max_depth)
7 prob ← "(" + left_expr + " " + operator + " " + right_expr + ")"
8 **return** prob

---

### A.2 COMBINED-PROBLEM INPUT SCENARIO

#### A.2.1 UQ AND SQ

For the two types of problems, our requirement for UQ is that QwQ-32B cannot solve it correctly, whereas for SQ, we require QwQ-32B to solve it correctly. For each of SQ and UQ, we manually selected one problem from Graph Theory(West et al., 2001) and verified that it meets these criteria on QwQ-32B. Below, we directly present both of these selected problems.

- UQ: An up/down labeling is a graceful labeling for which there exists a critical value a such that every edge joins vertices with labels above and below a. Prove that every caterpillar has an up/down labeling. Prove that the 7-vertex tree that is not a caterpillar has no up/down-labeling.

- SQ: The Petersen graph has even more symmetry than vertex-transitivity. Let $P = (u_0, u_1, u_2, u_3)$ and $Q = (v_0, v_1, v_2, v_3)$ be paths with three edges in the Petersen graph. Prove that there is exactly one automorphism of the Petersen graph that maps $u_i$ into $v_i$ for i = 0, 1, 2, 3.

#### A.2.2 INPUT MESSAGE TEMPLATE

This section we will provide the prompt of combined-problem input in Figure 5.

```
[{"role": "user",
"content": "Please address the following problems: Problem 1: "
+ SQ/UQ + "\nProblem 2: Calculate " + arithmetical problem}]
```

Figure 5: Combined-problem Message Template

#### A.2.3 TRUNCATING OR PADDING THE REASONING TRACE

In Section 3.2, to eliminate the confounding effect of context length, we normalize the reasoning context length in the combined SQ/UQ + Arith input setting. To preserve coherence, we perform adjustments at the *segment level*. Specifically, we first locate all occurrences of the keyword "Alternatively," which marks shifts in reasoning, and treat the tokens between two consecutive occurrences as a segment. As illustrated in Figure 6, for UQ + Arith inputs (which tend to be longer), we progressively remove segments from the beginning until the average length is shorter than that of SQ + Arith. Conversely, for SQ + Arith inputs (which are shorter), we duplicate the first segment and append it repeatedly until the average length exceeds that of UQ + Arith.

This procedure ensures that, when reasoning for *Problem 2* begins, the context lengths of UQ + Arith and SQ + Arith inputs are comparable, thereby controlling for the effect of context length.

```
702   vanilla context:
703   [segment 1] + 'Alternatively' +[segment 2] +'Alternatively' +
704   [segment 3] + 'Alternatively' +[segment 4] ...'Alternatively' +
705   [segment n] + [Reasoning trace of Problem 2]
706
707   Padding context:
708   [segment 1] + 'Alternatively' +[segment 2] +'Alternatively' +
709   [segment 2] + 'Alternatively' +[segment 2] ...(Repeat)...
710   'Alternatively' + [segment 3] + 'Alternatively' +[segment 4] ...
711   'Alternatively' + [segment n] + [Reasoning trace of Problem 2]
712
713   Truncating context:
714   [segment 1] + 'Alternatively' + [segment n-k] + 'Alternatively' +
715   [segment n-k+1] + ... 'Alternatively' +[segment n] +
      [Reasoning trace of Problem 2]
```

Figure 6: Truncating or Padding the Reasoning Trace

### A.3 MORE DETAILS ABOUT D-REFINE

In this section, we provide the formalized algorithm of D-Refine, which is designed to handle the more realistic setting where a single problem is given as input. The detailed execution procedure is presented in Algorithm 2.

---

**Algorithm 2** D-Refine

**Input:** Input problem $P$, trigger length $s_{len}$, maximum refine times $T_{max}$
**Output:** Final answer `ans`

9  Initialize context $C \leftarrow P$, refine count $t \leftarrow 0$
10 **while** *reasoning not finished* **do**
11   Generate next token $x$ based on context $C$
12   Append $x$ to $C$
13   **if** $length(C) \geq s_{len}$ **and** $t < T_{max}$ **then**
14    Extract drafting trace $D$ from $C$
15    Summarize $D$ into a concise structured form $S$
16    Update context $C \leftarrow P + S$
17    $t \leftarrow t + 1$
18 **return** final answer `ans`

---

#### A.3.1 PROMPT FOR REFINEMENT

To enhance reproducibility, we provide the exact prompt used during the refinement phase. The goal of refinement is to make the drafting trace more concise and structured, while removing information irrelevant for subsequent reasoning. Notably, the LLM is not required to verify the correctness of prior reasoning steps. The prompt is shown in Code 1.

#### A.3.2 PROMPT FOR REASONING

In addition to the refinement prompt, we also provide the prompt used during the reasoning phase. This prompt is applied both in the *Direct Reasoning* baseline and in the reasoning stage of *D-Refine*. The detailed template is shown in Code 3.

Listing 1: Prompt for refinement

```
THE_REFINEMENT_PROMPT_TEMPLATE = """
You are given:
```

```
A math problem.
A long, natural-language reasoning process generated by an LLM during
extended thinking (the reasoning may be incomplete, contain mistakes, or
have self-contradictions).
Your task is to:
Organize the original lengthy reasoning into concise notes.
Remove any content that is explicitly discarded or identified as wrong by
 the original reasoning.
Remove repetitive or redundant statements.
Do NOT verify or correct the logic or mathematical correctness. If the
original reasoning contains errors that are not explicitly rejected
within the text, keep them.
Make the final output read like neat, well-structured notes from human
scratch work.

Follow these steps:

Read the entire provided reasoning process.
Identify and eliminate any portion the reasoning itself disqualified or
labeled as incorrect.
Eliminate repeated or obviously redundant statements.
Preserve the remaining thoughts in a concise, organized, note-style
summary.
Do not add any new information or corrections.
Return your final summary in a format that reads naturally, something
like a cleaned-up set of notes.
Your output should look like:

<organized_notes>
(Here you provide the structured summary of the thought process, keeping
any errors that are not explicitly rejected, but removing redundancies
and self-contradicted parts.)
</organized_notes>

Please provide your organized note-style reasoning based on these
instructions.
"""

USER_INPUT_FORMAT_TEMPLATE = """
Below is the input:

Problem:
{PROBLEM HERE}

Original Reasoning:
{REASONING HERE}
"""
```

Listing 2: Prompt for Reasoning

```
THE_REASONING_PROMPT_TEMPLATE_BASELINE = """
Please address the following problem: [Problem]
"""

THE_REASONING_PROMPT_TEMPLATE_D_REFINE = """
Please address the following problem: [Problem]
Based on the following organized notes from your previous thoughts,
continue your reasoning.
 <organized_notes>\n[NOTES]n</organized_notes>
Continue your thinking now.
"""
```

## A.4 CASE STUDY OF D-REFINE

In this section, we provide examples of refinement, including both SQ/UQ + Arithmetic input scenarios and single-question input scenarios. Due to space constraints—since a full case reasoning process typically exceeds 10k tokens—the complete cases are compiled into a JSON file, while a small subset of examples is included in the supplementary material.

Listing 3: Prompt for Reasoning

```
#QwQ-32B D-Refine maximum draft length set: 9k, number of refinement: 3

Problem="""
Calculate (((((-0.66 + -9.05) + (1.56 + -7.07)) - ((-9.2 - -7.76) +
(-3.92 - 9.83))) + (((-9.05 - 8.19) - (-6.47 - 0.51)) + ((-4.82 - 8.93)
+ (-5.37 + -7.88)))) - (((((-6.95 + 3.33) - (9.83 - 4.23)) + ((0.17 +
-3.42) - (6.07 + -2.11))) + (((-9.09 + 9.1) + (9.4 + 8.04)) - ((7.27 +
-2.47) - (-4.05 - -7.25)))))
"""

Refinement_noted="""
<organized_notes> 1. **First Chunk**: (((-3.93 + 1.57) + -8.64) - (-6.98
+ (1.09 + -4.63))) - -3.93 + 1.57 = -2.36 - -2.36 + -8.64 = -11.00 - 1.09
 + -4.63 = -3.54 -> -6.98 + (-3.54) = -10.52 -> -(-10.52) = +10.52 -
Result: -11.00 - (-10.52) = -0.48 2. **Second Chunk**: (((-3.28 + 3.57) -
 (-2.67 -8.41)) - ((0.47 -8.62) - (5.72 + -6.47))) - -3.28 + 3.57 = 0.29
-> 0.29 - (-11.08) = 11.37 - 0.47 -8.62 = -8.15 -> 5.72 + -6.47 = -0.75
-> -8.15 - (-0.75) = -7.40 - Result: 11.37 - (-7.40) = 18.77 - Cumulative
 Total: -0.48 + 18.77 = 18.29 3. **Third Chunk**: ((((6.36 + -4.29) -
-6.65) - ((4.8 +1.74) - (-0.08 + -3.18))) - (((9.94 +4.18) - (-9.74
+6.36)) - (-4.5 - (-8.83 + -1.28)))) - 6.36 + -4.29 = 2.07 -> 2.07 -
-6.65 = 8.72 - 4.8 +1.74 = 6.54 -> -0.08 + -3.18 = -3.26 -> 6.54 -
(-3.26) = 9.80 -> 8.72 - 9.80 = -1.08 - 9.94 +4.18 =14.12 -> -9.74 +6.36
=-3.38 ->14.12 - (-3.38) =17.50 - 4.5 - (-10.11) =5.61 ->17.50 -5.61
=11.89 ->-1.08 -11.89 =-12.97 - Result: -12.97 - Cumulative Total: 18.29
+ (-12.97) =5.32 4. **Fourth Chunk**: (((((7.29 -4.96) - -6.62) - ((5.74
-8.98) - (-6.97 -9.21))) + (((9.11 - -5.31) -2.56) + ((2.31 +2.51) -
-9.25))) + ((((3.48 - -6.84) - (-1.66 - -5.29)) - ((-5.01 -0.69) + (9.99 -
 -7.01))) - (((5.99 + -2.6) -1.16) + ((9.66 + -0.73) -7.77)))) - 7.29
-4.96 =2.33 ->2.33 - -6.62 =8.95 - 5.74 -8.98 =-3.24 ->-6.97 -9.21
=-16.18 ->-3.24 - (-16.18) =12.94 ->8.95 -12.94 =-3.99 - 9.11 - -5.31
=14.42 ->14.42 -2.56 =11.86 - 2.31 +2.51 =4.82 ->4.82 - -9.25 =14.07
->11.86 +14.07 =25.93 - 3.48 - -6.84 =10.32 ->-1.66 - -5.29 =3.63 ->10.32
 -3.63 =6.69 - -5.01 -0.69 =-5.70 ->9.99 - -7.01 =17.00 ->-5.70 +17.00
=11.30 ->6.69 -11.30 =-4.61 - 5.99 + -2.6 =3.39 ->3.39 -1.16 =2.23 - 9.66
+ -0.73 =8.93 ->8.93 -7.77 =1.16 ->2.23 +1.16 =3.39 ->-4.61 -3.39 =-8.00
 - Result: (-3.99 +25.93) + (-8.00) =13.94 - Cumulative Total:5.32 +13.94
 =19.26 5. **Fifth Chunk (Subtracted)**: - ((((((-4.91 + -2.47) - (-8.13
-0.89)) + ((3.32 -1.97) +3.68)) - ((-3.81 + (7.53 +7.22)) - ((-8.35
-2.02) -6.56))) - (((4.04 +8.09) + ((-6.45 -5.94) +9.14)) - (((6.22
+2.55) - (-4.62 -8.54)) + ((2.37 +2.2) - (9.23 +2.24))))) + (((((-1.99 -
(3.51 + -2.51)) + ((8.49 +1.7) - -7.37)) + ((-7.52 - (7.52 + -6.06)) +
((6.78 -1.81) + (-5.9 -1.27)))) + ((((7.03 + -1.77) - (-2.84 +5.79)) +
((-3.84 +6.89) - -7.04)) - ((-3.38 +3.61) - (0.49 + (-6.78 -7.75))))))) -
 Key Steps: - PartA: -21.20 - PartB: -6.15 - Final: - [ (-21.20 -6.15) +
... ] -> simplified to -4.21 **Final Total**: 19.26 (cumulative before
last chunk) - (-4.21) = **23.47** </organized_notes>

#QwQ-32B D-Refine maximum draft length set: 6k, number of refinement: 3
Problem="""
A list of positive integers has the following properties: \bullet The sum
 of the items in the list is 30. \bullet The unique mode of the list is 9.
 \bullet The median of the list is a positive integer that does not
appear in the list itself. Find the sum of the squares of all the items
in the list.
"""
```

```
Refinement_noted="""
 <organized_notes> **Problem Requirements:** - Sum of list = 30 - Unique
 mode = 9 (appears more than any other number) - Median is an integer not
  in the list (implies even number of elements) **Analysis for n=4 (even
 length):** - Let list = [a, b, 9, 9] (sorted, with two 9s as mode). -
 Sum: \(a + b + 9 + 9 = 30 \Rightarrow a + b = 12\). - Median = \(\frac{b
  + 9}{2}\) must be integer not in the list. - Valid pairs (a, b) with \(
 a \leq b \leq 9\): - \(a=5, b=7\): Median = \(\frac{7+9}{2}=8\) (not in
 list). - Valid list: [5, 7, 9, 9]. - Sum: \(5 + 7 + 9 + 9 = 30\). - Mode:
  9 (appears twice, others once). - Median 8 not in list. **Sum of
 squares for [5,7,9,9]:** \(5^2 + 7^2 + 9^2 + 9^2 = 25 + 49 + 81 + 81 =
 236\). **n=6 analysis (discarded):** - Three 9s require remaining
 numbers to sum to 3 (impossible without duplicate modes). - Two 9s lead
 to non-integer median or duplicate modes in valid configurations. **
 Conclusion:** Only valid list is [5,7,9,9]. Sum of squares = 236. </
 organized_notes>
 """

#Llama D-Refine maximum draft length set: 3k, number of refinement: 3
Problem="""
Let $\triangle ABC$ be a right triangle with $\angle A = 90^\circ$ and
$BC = 38$. There exist points $K$ and $L$ inside the triangle such that
$$AK = AL = BK = CL = KL = 14.$$ The area of the quadrilateral $BKLC$ can
 be expressed as $n\sqrt{3}$ for some positive integer $n$. Find $n$.
 """

Refinement_noted="""
 <organized_notes> The problem involves a right triangle ABC with a right
  angle at A and hypotenuse BC = 38. Points K and L inside the triangle
 satisfy AK = AL = BK = CL = KL = 14. We need to find the area of
 quadrilateral BKLC, expressed as nsqrt(3), and determine n. By placing
 the triangle in a coordinate system with A at the origin, B at (c,0),
 and C at (0,b), we derive the following key equations: 1. \( c^2 + b^2 =
  38^2 = 1444 \) 2. Points K and L lie on circles centered at A with
 radius 14, leading to their coordinates satisfying: - \( x_1^2 + y_1^2 =
  196 \) - \( x_2^2 + y_2^2 = 196 \) 3. Distances from B to K and C to L
 give: - \( (x_1 - c)^2 + y_1^2 = 196 \) - \( x_2^2 + (y_2 - b)^2 = 196 \)
  4. The distance between K and L is 14: - \( (x_2 - x_1)^2 + (y_2 - y_1)
 ^2 = 196 \) Solving these equations, we find that \( x_1 = c/2 \) and \(
  y_2 = b/2 \). Substituting back, we derive relationships involving c
 and b, leading to the equation \( c x_2 + b y_1 = 196 \). However,
 solving for c and b directly is complex due to the transcendental nature
  of the resulting equations. Considering the problem's requirement for
 the area of BKLC to be expressed as \( n\sqrt{3} \), we explore the
 possibility of the triangle being a 30-60-90 triangle, but this leads to
  inconsistencies. Ultimately, recognizing the complexity, we conclude
 that the area of BKLC is \( 98\sqrt{3} \), thus \( n = 98 \). **Answer:**
  The value of \( n \) is \boxed{98}. </organized_notes>
 """
```

## A.5 The Use of Large Language Models

Large language models were primarily employed to improve the paper's grammatical correctness and to enhance the fluency and rigor of the text.

