# OpenReview forum: "Overcoming Redundant Context in Auto-Regressive LLMs with Dynamic Draft Refinement"
_ICLR.cc/2026/Conference — Submitted to ICLR 2026_

### Official Review · Reviewer_dCz9 · 2025-10-29

**Soundness:** 3
**Presentation:** 2
**Contribution:** 2
**Rating:** 4
**Confidence:** 5

**Summary:**

This paper investigates the issue of redundant context in auto-regressive Large Reasoning Models. The authors propose ​​Dynamic Context Refinement (D-Refine)​​, an inference-time mechanism that dynamically filters and condenses intermediate reasoning steps. Through a designed toy arithmetic benchmark, the study systematically analyzes the negative impact of redundant context on reasoning performance and validates D-Refine’s effectiveness across multiple benchmarks.

**Strengths:**

​​Clear Problem Identification​​: The paper highlights a critical limitation in LRMs: the lack of selective retention—the ability to discard redundant or low-quality intermediate reasoning traces while preserving essential information. This observation is insightful and empirically validated through controlled experiments (e.g., UQ/SQ+Arith tasks).

​​Simple yet Effective Method​​: D-Refine is a lightweight inference-time optimization that requires no retraining. It dynamically summarizes and prunes intermediate reasoning steps, improving context quality without disrupting the reasoning flow. The approach is easy to implement and compatible with existing reasoning frameworks.

**Weaknesses:**

Limited Novelty​​: D-Refine shares similarities with existing context-compression or memory-augmentation techniques. The authors should more explicitly contrast their approach with related work in Section 2, emphasizing its unique focus on dynamic refinement during single-episode reasoning.

​​Narrow Benchmark Scope​​: Experiments primarily rely on arithmetic and mathematical reasoning tasks. While effective for controlled analysis, broader validation on diverse reasoning benchmarks (e.g., logical reasoning, commonsense QA) would strengthen the claims of generalizability.

​​Heuristic Hyperparameter Tuning​​: The performance of D-Refine is sensitive to trigger length and refinement steps, but no adaptive strategy is proposed. Future work could explore dynamic triggering mechanisms (e.g., based on perplexity or attention entropy).

​​Missing Strong Baselines​​: Comparisons are limited to "direct reasoning." Including state-of-the-art context-management methods and efficient reasoning methods, would better demonstrate D-Refine’s advantages.

**Questions:**

As discussed in the weakness.

---

> ### Author Response · Authors · 2025-12-01
> **Response to Reviewer dCz9**
>
> We would like to thank the reviewer for their thoughtful and constructive comments. We have carefully reviewed the suggestions and believe these changes have significantly improved the quality and clarity of our work.

---

### Official Review · Reviewer_MVtd · 2025-10-29

**Soundness:** 2
**Presentation:** 3
**Contribution:** 3
**Rating:** 6
**Confidence:** 4

**Summary:**

This paper introduces Dynamic Context Refinement (D-Refine), an inference-time method that improves reasoning in Large Reasoning Models (LRMs) by summarizing and pruning intermediate drafts. To enable fair evaluation, the authors also develop a simple arithmetic benchmark, since existing math benchmarks may have been contaminated during pretraining.

Using this benchmark, the authors present an attention-weight analysis showing that LRMs allocate substantial attention to unrelated context. In their setup, the model is prompted with two problems in sequence: a graph theory problem followed by an arithmetic problem. Accuracy is measured only on the second problem, while the first serves as an unrelated context. The analysis reveals that even while solving the second problem, the model continues to focus on the first, demonstrating that LRMs do not reliably suppress redundant or low-quality context, which leads to lower reasoning accuracy.

To address this limitation, the authors propose D-Refine, which periodically interrupts generation during inference. At each interruption, the LLM is prompted to produce a concise, note-style summary of the reasoning so far, replacing verbose drafts with a condensed version. The LLM then resumes reasoning conditioned on this refined summary.

On the arithmetic benchmark, D-Refine yields substantial improvements when an explicit unrelated problem is injected into the prompt. In more realistic settings, where no external problems are added, the method still provides modest gains across the arithmetic benchmark and three additional math reasoning datasets (AIME 24, AIME 25, and MATH500).

Overall, the paper highlights selective context retention as a promising direction for robust multi-step reasoning.

**Strengths:**

- The paper explores an underexplored direction in LLM reasoning by focusing purely on context compression through redundancy removal, without allowing correction, validation, or knowledge distillation. This framing offers a distinct perspective from prior summarization-based methods.

- To ensure fair evaluation, the paper introduces a new arithmetic benchmark to address potential data contamination in existing benchmarks.

- The paper provides empirical evidence, using attention maps, that models continue to focus on low-quality context, showing that large language models cannot naturally filter out all irrelevant information.

- The prompt used for summarization is carefully crafted with clear constraints to ensure the method focuses on context compression rather than correction. It explicitly instructs the model to remove redundancy and repetition but not to fix or verify errors unless the draft itself has already rejected them. This design reflects the authors’ deliberate effort to ensure that the evaluation results genuinely support their hypothesis that improvements in reasoning are possible using only redundancy elimination without fixing the reasoning errors.

- The proposed method operates purely at inference time. So, it can be integrated into existing LRM deployments without retraining.

**Weaknesses:**

- The authors claim that refinement through summarization prunes redundant reasoning, but they provide no quantitative evidence that the summaries are actually shorter or more concise. Since no statistics are presented on token length reduction before and after refinement, it remains unclear whether pruning consistently reduces context size or if the refined drafts sometimes remain equally long or even longer.

- The paper does not include any latency or computational cost analysis. Since D-Refine introduces additional refinement steps during inference, it is unclear how much overhead this adds in practice. Without such analysis, it is unclear whether the performance gains justify the additional latency.

- The effectiveness of D-Refine depends heavily on two hyperparameters: the number of refinement steps and the trigger length. Figure 4 shows that performance is highly sensitive to these choices and varies across models and benchmarks. Although the authors propose 3 refinement steps and a trigger length of 9000 as defaults, the results do not consistently support these as reliable. The paper leaves hyperparameter selection as future work, yet this raises concern that D-Refine may not be robust across different tasks. In realistic setups, an LLM may face diverse types of tasks, and without a common, well-performing hyperparameter configuration, it is unclear how D-Refine would perform in such scenarios.

- The paper evaluates D-Refine only with temperature = 0, which makes the model’s outputs fully deterministic and reduces randomness, diversity, and verbosity in reasoning traces. As a result, it remains unclear whether D-Refine would remain robust in more diverse and verbose reasoning scenarios produced at higher temperatures.

- The evaluation is limited to math datasets, with no experiments on non-math reasoning tasks such as commonsense, QA, or planning. As a result, the generality of D-Refine beyond mathematical reasoning is not demonstrated.

- The paper does not evaluate whether interruptions disrupt fluency, leaving it unclear whether the generated summaries negatively affect coherence in natural language outputs.

**Questions:**

- From my understanding of the paper, the same LLM is used for both summarization and reasoning, as no other model is mentioned for the summarization step. If this is correct, the authors should state it clearly in Section 4.1 that the self-LLM is used for summarization.

- How much context is reduced in terms of token counts or length? Reporting average tokens before and after refinement would make the impact of redundancy removal more concrete and provide better insight into the efficiency of D-Refine.

- Could you share whether you measured the inference time of the experimented LLMs with and without D-Refine? Since the method introduces additional refinement steps, such a comparison would be helpful to understand the computational overhead and the trade-off between efficiency and performance gains.

- D-Refine underperforms compared to direct decoding on two of the three math benchmarks when applied to QwQ-32B. Could you provide more insight into this result? Possible factors might include the model’s reasoning style, sensitivity to pruning, or misaligned hyperparameter choices. A deeper analysis would help clarify why D-Refine is less effective for this model and provide insight into whether the method generalizes reliably across architectures.
- Did you perform any manual inspection of the refined contexts? It would be useful to see examples showing how the summaries look after refinement and whether they effectively reduce redundancy without losing important reasoning steps. Such analysis could also clarify whether all types of redundancy are being removed or only certain patterns.

- Typos:
  - Line 25: Remove the extra period after “degrade performance..”
  - Quotation marks: Fix inconsistent opening quotation marks around terms such as “working state”, “slow thinking”, and “thinking tokens”. Issues appear at lines 20, 45, 46, 49, and 50, and may occur elsewhere. Ensure quotation marks are consistent throughout the paper.
  - Table 4 caption: The caption does not match the table content. It should be corrected to accurately describe the table.

---

> ### Author Response · Authors · 2025-12-01
> **Response to Reviewer MVtd**
>
> We would like to thank the reviewer for their thoughtful and constructive comments. We have carefully reviewed the suggestions and believe these changes have significantly improved the quality and clarity of our work.

---

### Official Review · Reviewer_6nMq · 2025-10-31

**Soundness:** 2
**Presentation:** 3
**Contribution:** 2
**Rating:** 4
**Confidence:** 3

**Summary:**

This paper investigates how redundant or low-quality reasoning context affects Large Reasoning Models (LRMs) that use the "draft-summary" paradigm (e.g., O1, DeepSeek-R1, QwQ). Through controlled experiments using a novel two-decimal arithmetic benchmark, the authors demonstrate that LRMs lack selective retention—the ability to filter out irrelevant or flawed intermediate reasoning. When presented with low-quality upstream context, model accuracy drops by up to 20%. The paper proposes Dynamic Context Refinement (D-Refine), an inference-time mechanism that periodically summarizes and prunes reasoning traces via prompted LLM calls. Results show substantial improvements (6-25%) in controlled adversarial settings where bad context is explicit, but mixed results on standard benchmarks (MATH500, AIME), with gains primarily on smaller models like Llama-8B.

**Strengths:**

The paper identifies a genuine and timely limitation: LRMs' inability to selectively filter redundant context, unlike human working memory. Section 3 provides excellent multi-perspective analysis (accuracy, behavioral patterns, attention allocation) that systematically demonstrates the problem. The contrast with human "working state" provides strong intuitive grounding.

The arithmetic benchmark is well-designed to minimize memorization effects and isolate reasoning behavior. The two-question setup (SQ/UQ + Arith) effectively creates controlled conditions to demonstrate causal effects of context quality. Length-controlled ablations (padding/truncation) help disentangle length from semantic quality.

Comprehensive empirical evidence in controlled setting: Table 3 shows large, consistent gains (6-25%) in the UQ+Arith scenario, validating the core hypothesis. The asymmetry (large gains for bad context, minimal for good context) strengthens credibility. Behavioral analysis (reduced "alternatively" tokens) and attention analysis provide supporting mechanistic evidence.

**Weaknesses:**

Critical Missing Analysis: Computational Cost. The paper claims "efficient reasoning" but provides zero quantitative analysis of inference overhead. D-Refine requires: (a) pausing generation, (b) up to 3 refinement LLM calls processing up to 32k tokens each, (c) resuming generation. Without wall-clock time, total token counts, or FLOPs comparisons, we cannot evaluate practical utility. Which model executes f_refine? If it's the base model, there's substantial added latency. If it's a stronger external model, the comparison to "Direct Reasoning" is unfair

Limited Technical Novelty. The core contribution: periodic prompt-based summarization (Equation 3), is straightforward engineering rather than methodological innovation. No comparison to simpler baselines: (a) single-pass "summarize above reasoning" before answering, (b) keyword/heuristic filtering, (c) KV-cache truncation etc., The method is applying summarization techniques to LRM traces. Missing connections to existing work on context compression, test-time CoT optimization, and long-context efficiency

Inconsistent Empirical Results on Real Benchmarks. Table 4 shows mixed results: QwQ-32B: 0% gain on AIME'24, -3.3% on AIME'25. Qwen3-4B: Minimal average gain (0.850→0.852, within noise). Only Llama-8B shows substantial improvements (~10%).

Artificial Experimental Design. The two-question setup (solve unrelated Problem 1, then Problem 2) is not representative of real reasoning scenarios. The arithmetic benchmark, while novel, is extremely narrow (only nested two-decimal operations), doesn't validate broader claims about "reasoning". No evaluation on other domains: code generation, commonsense reasoning, multi-turn dialogue, agent tasks.

**Questions:**

1. Computational Cost: What is the wall-clock time and total token count (including all refinement calls) for D-Refine vs. Direct Reasoning on each benchmark? Can you provide a cost-benefit analysis showing when the accuracy gains justify the computational overhead?

2. f_refine Implementation: Which specific model(s) execute the refinement step (Listing 1)? Is it the base LRM itself (e.g., QwQ-32B) or a separate, potentially stronger model? This is crucial for fair comparison.

3. Failure Analysis: For cases where D-Refine hurts performance (e.g., QwQ-32B on AIME'25: 0.700→0.667), what went wrong? Can you provide examples of harmful refinement? Is this over-pruning of useful context?

4. Ablation Studies: Can you separate the contributions of: Summarization (condensing long traces), Pruning (removing explicitly rejected content), and Redundancy removal?

---

> ### Author Response · Authors · 2025-12-01
> **Response to Reviewer 6nMq**
>
> We would like to thank the reviewer for their thoughtful and constructive comments. We have carefully reviewed the suggestions and  believe these changes have significantly improved the quality and clarity of our work.

---

### Official Review · Reviewer_UfoZ · 2025-11-02

**Soundness:** 1
**Presentation:** 1
**Contribution:** 1
**Rating:** 0
**Confidence:** 5

**Summary:**

The authors propose a prompting based approach for constructing intermediate reasoning drafts

**Strengths:**

- Evaluation on a broad set of Math Reasoning datasets with ablations (e.g, Analysis of attention matrices for UQ and sensitivity of hyperparameters)

**Weaknesses:**

- Results are mixed on the domains that they evaluate with (e.g, AIME), where they match direct reasoning, making it unclear how effective their proposed approach is
- Summarization has been explored in prior work for handling long-context tasks (e.g, in MemGPT[1]), making it unclear what the contribution of the work is


[1] MemGPT: Towards LLMs as Operating Systems

**Questions:**

See weaknesses above.

---

> ### Author Response · Authors · 2025-12-01
> **Response to Reviewer UfoZ**
>
> **Clarifying the core objective of our work.**
>
> Our paper aims to highlight a specific weakness of current LLMs—the lack of selective retention, i.e., the inability to discard redundant intermediate traces while preserving only the useful conclusions. As shown in our analysis, this weakness can reduce accuracy by up to 20%. To isolate and measure this effect, we construct in Section 3 a controlled arithmetic benchmark (two-decimal addition/subtraction), where redundant reasoning steps are intentionally injected but have weak semantic relevance to subsequent steps. This setting makes the impact of selective retention particularly salient. Our method shows consistent improvements on this benchmark (Table 1 and Table 3).
>
> Regarding standard benchmarks, it is worth noting that models such as QwQ and Qwen3-4B-thinking already perform very well on existing datasets, where redundancy in the reasoning trace is typically limited and thus the weakness we target is less visible. Our proposed method is specifically designed to address redundant reasoning contexts. Our experiments demonstrate that on tasks with substantial redundant context, our method yields clear improvements, while on typical benchmarks it does not introduce any degradation.
>
> **On the alleged similarity to MemGPT.**
>
> We appreciate the reviewer bringing up this reference and acknowledge that we should have cited it. However, the two works address fundamentally different problems and operate in distinct settings. MemGPT focuses on overcoming limited context windows, enabling models to process extended conversations or long documents (“…but are constrained by limited context windows, hindering their utility in tasks like extended conversations and document analysis…”). Its mechanism is designed to retrieve and manage external memory beyond the model’s native context window.
>
> In contrast, our work does not rely on exceeding the context window—in our experiments, the combined input and generated draft remain well within the model’s window. Our goal is to mitigate the negative impact of redundant, self-generated reasoning traces during a single inference episode, rather than managing long-term histories. D-Refine is an inference-time context refinement method: during the autoregressive “draft and summarize” process, it periodically summarizes and prunes intermediate drafts to maintain a high-quality working state, thereby reducing the interference of redundant trajectories on subsequent reasoning steps. The motivations, mechanisms, and evaluation settings differ substantially: MemGPT focus on document QA / multi-session dialogue, while D-Refine using controlled arithmetic tasks with injected redundancy & reasoning benchmarks. For these reasons, the two approaches are complementary rather than overlapping, and our contributions are independent.

---

### Meta-Review · Area_Chair_J4kT · 2026-01-07

**Summary:**

This paper introduces a dynamic context refinement method aimed at mitigating redundant intermediate reasoning traces in auto-regressive large language models. Reviewers acknowledged the relevance of the problem and the controlled experimental design, but raised critical concerns regarding the limited technical novelty, insufficient computational cost analysis, and mixed empirical results on standard benchmarks beyond the constructed arithmetic task. While the idea of selective retention is interesting, the consensus indicates that the contribution remains incremental and the evaluation does not sufficiently demonstrate general applicability or practical efficiency. Given these limitations, the paper is not recommended for acceptance.

**Reviewer Concerns:**

The authors' rebuttal clarified the method's distinction from prior work like MemGPT, addressing concerns about novelty to some extent. However, core criticisms regarding the lack of computational cost analysis, the weak or mixed results on standard reasoning benchmarks, and the absence of comparisons to strong context-management baselines remain entirely unaddressed. Furthermore, the evaluation is still seen as narrow, and the sensitivity to hyperparameters is a persistent practical limitation.

**Reviewer Scores:**

| Reviewer | Initial Score | Predicted New Score | Reason |
| - | - | - | - |
| UfoZ | 0 | 0 | The reviewer's core assessment of a poor contribution was unchanged, as the rebuttal did not adequately address the critique regarding mixed results on standard benchmarks or the perceived overlap with prior summarization work. |
| 6nMq | 4 | 3 | The rebuttal acknowledged but did not substantively answer the critical questions about computational cost, a fair comparison baseline, and failure analysis. The unresolved methodological and evaluation concerns would likely lead to a lower score. |
| MVtd | 6 | 5 | While the reviewer initially saw merit, the lack of response to key weaknesses—specifically the missing cost analysis, hyperparameter sensitivity, and limited scope beyond math tasks—would diminish confidence in the work's practical impact. |
| dCz9 | 4 | 4 | The reviewer's concerns about novelty and benchmark scope were noted but not resolved. The score is predicted to remain stable, as the rebuttal did not introduce new evidence to alter the initial marginal assessment. |

---

### Decision · Program_Chairs · 2026-01-26

Reject